# OpenReview forum: "Hybrid Architectures for Language Models: Systematic Analysis and Design Insights"
_ICLR.cc/2026/Conference — Submitted to ICLR 2026_

### Official Review · Reviewer_Lhf2 · 2025-10-22

**Soundness:** 3
**Presentation:** 2
**Contribution:** 2
**Rating:** 6
**Confidence:** 2

**Summary:**

This paper presents a hybrid architecture combining Transformer and Mamba modules to enhance inference speed and computational efficiency. The idea is novel and relevant, showing potential in bridging attention-based and state-space models. However, the paper lacks clarity on when and how each module is applied, as well as theoretical justification for their integration. The model design and experimental details are insufficiently explained, and the absence of thorough ablation and generalization studies limits the strength of the empirical claims. Overall, the contribution is promising but not yet fully convincing.

**Strengths:**

The paper introduces a novel combination of Transformer and Mamba modules, aiming to exploit their complementary strengths in sequence modeling and efficiency.

The work addresses a relevant challenge in modern deep learning — improving inference speed and memory efficiency without significantly compromising accuracy.

Preliminary results demonstrate that the proposed hybrid architecture can outperform standard Transformer baselines on certain tasks, suggesting its promise for long-sequence or resource-constrained applications.

**Weaknesses:**

The paper does not clearly explain the theoretical motivation for combining Transformer and Mamba modules, nor provide criteria for when each should be activated.

Key implementation aspects such as module interaction, feature fusion, and parameter sharing are not well described, which limits reproducibility.

The evaluation is narrow, lacking ablation studies, robustness analysis, and tests on diverse datasets to support the claimed generalization and efficiency improvements.

**Questions:**

1. How does the model determine when to use the Transformer module versus the Mamba module? Is there a data-driven or theoretical basis for this choice?
﻿
2. Are the integration strategies between modules such as information flow, gating mechanisms, or parameter sharing fixed or adaptive during training?
﻿
3. Could the authors provide ablation studies to quantify the contribution of each module and demonstrate the stability of the hybrid architecture across tasks?
﻿
4. How does the proposed hybrid approach generalize to different sequence lengths and domains beyond the tested benchmarks?

---

> ### Author Response · Authors · 2025-11-24
>
> We thank the reviewer for their constructive feedback. We are pleased that they recognized the novelty of our hybrid architecture in leveraging complementary strengths to balance efficiency with accuracy, highlighting its potential for long-sequence and resource-constrained applications. We provide responses to the reviewer’s specific questions and concerns (quoted) below.
>
> ---
>
> > **[W1] Lack of motivations or criteria for combining Transformer and Mamba modules.**
>
> **[A]** We appreciate the feedback. We believe our ablation study on block positioning gave some hints which modules should be activated at specific depths. Moreover, attention score analysis provided evidence that using Transformer components in early layers—where they tend to generate 'Bag-of-Vectors' representations by aggregating global context [1, 2]—can result in blurred context initialization, which is suboptimal when combined with Mamba.
>
> To further investigate the optimal activation patterns, we conducted additional experiments on the intra-hybrid architecture during this revision (see **Appendix J**). We pre-trained a 350M-parameter model from scratch where every layer consists of an intra-hybrid block. In this setup, we evenly divided the heads, applied Group Normalization to each module's output, and combined them via a weighted sum scaled by $\alpha$ and $1-\alpha$. Here, $\alpha$ serves as a quantitative proxy for each module's contribution to next-token prediction.
>
> Based on this, we performed two analyses and offered the following insights into the roles and contributions of the two modules.
>
> * **Layer-wise Analysis (Figure 8):** Consistent with our ablation studies, the model assigns higher weights to the Mamba component in earlier layers (particularly the first) to optimize performance, while heavily favoring the Transformer component in the middle layers (specifically layers 6 and 7).
>
> * **Token-wise Analysis (Table 12):** Visualization reveals distinct functional roles. The Transformer component is utilized more heavily at sentence boundaries (where new information is introduced) and at middle depths for retrieving key context words, numerical values, and complex information. In contrast, Mamba is more active during sub-word prediction (completing tokens split by the tokenizer), sentence endings, and for tasks requiring primarily local information or simple predictions.
>
>
> ---
>
> > **[W2] Not well described key implementation aspects.**
>
> **[A]** We apologize for the lack of detail regarding the key aspects of our intra-layer hybrid architecture. To address this, we have updated **Appendix E** to elaborate on the specific design axes and variants explored, clarifying the column definitions in **Table 5**. We also have included a more comprehensive set of ablation results in **Appendix I**. We encourage the reviewer to refer to these updated sections for further details.
>
> ---
>
> > **[W3] Narrow scope of evaluation.**
>
> **[A]** We believe that our extensive ablation studies, conducted across language modeling and two distinct retrieval benchmarks, provide strong evidence for the generalizability of our claims. Nevertheless, we acknowledge that extending the evaluation scope further would strengthen our arguments.
>
> To this end, we are currently assessing fine-tuning performance on downstream tasks such as QA, summarization, and math reasoning. We note that given the limited pre-training scale (60B tokens), the model naturally lacks robust zero-shot capabilities for complex tasks like math and coding; therefore, we prioritize fine-tuning results to properly evaluate its adaptability.
>
> ---
>
> > **[Q1] How does the model determine when to use one of the modules?**
>
> **[A]** It is inherently challenging to establish a rigorous theoretical basis for such hybrid architectures, given the black-box nature of combining two distinct computational primitives. However, we were able to investigate the rationale behind module selection from a data-driven perspective.
>
> For instance, we derived insights into block positioning through ablation studies and investigated supporting evidence via attention score and layer-wise contribution analyses. It appears that, to optimize NLL, the hybrid model effectively assigns optimized roles to each component through extensive training on large-scale data.

---

> > ### Author Response · Authors · 2025-11-24
> >
> > > **[Q2] Are the integration strategies fixed or adaptive during training?**
> >
> > **[A]** In our experiments, all integration strategies were fixed prior to training. Our primary objective was to evaluate the final performance of each variant, thereby providing valuable design insights for researchers and developers constructing foundation models.
> >
> > However, learnable components do also exist within these variants. For instance, parameters such as learnable scaling factors can adaptively regulate the contribution (and information flow) of each module's output during training.
> >
> > We believe that dynamically identifying the optimal configuration from a candidate pool during the training process represents an interesting research direction. Works such as STAR [3] and Composer [4] have explored neural architecture search (NAS) within the context of inter-layer hybridization. It would be particularly interesting to investigate whether the observations we derived through comprehensive trial-and-error align with the solutions discovered by such automated search processes.
> >
> > ---
> >
> > > **[Q3] Ablation studies to quantify the contribution of each module?**
> >
> > **[A]** As detailed above, the newly added token-wise contribution analysis in **Appendix J** allows us to observe specific instances where each module is preferentially utilized. Specifically, the model appears to leverage Global Attention when retrieval capabilities or the introduction of new information are required, whereas Mamba is predominantly engaged in contexts necessitating local information processing.
> >
> > ---
> >
> > > **[Q4] Generalizability to different sequence lengths and domains.**
> >
> > **[A]** Experiments on various downstream tasks across different domains are currently in progress. We will update the results as soon as they become available.
> >
> > ---
> >
> > **References:**
> >
> > [1] Clark, et al. “What Does BERTLookAt? AnAnalysis of BERT’s Attention”. ACL 2019.
> > [2] Voita, et al. “Analyzing Multi-Head Self-Attention: Specialized Heads Do the Heavy Lifting, the Rest Can Be Pruned”. ACL 2019.
> > [3] Thomas, et al. “Star: Synthesis of tailored architectures”. ICLR 2025.
> > [4] Acun, et al. “Composer: ASearchFrameworkforHybrid Neural Architecture Design”. arXiv 2025.

---

> > > ### Comment · Reviewer_Lhf2 · 2025-11-28
> > >
> > > Thank you for the authors' response. The detailed reply has resolved most of my concerns. However, considering the overall contribution and novelty of the paper, I have decided to maintain my original score, as it is already a positive score.

---

> > > > ### Author Response · Authors · 2025-11-30
> > > >
> > > > We sincerely appreciate your valuable time and constructive comments, which have helped improve our paper. We will revise the manuscript to better emphasize our contributions.

---

### Official Review · Reviewer_xUqE · 2025-10-29

**Soundness:** 2
**Presentation:** 3
**Contribution:** 1
**Rating:** 2
**Confidence:** 2

**Summary:**

The paper studies hybrid language model architectures that mix Transformer attention with state-space models (SSMs), especially Mamba. It compares two ways to combine them: inter-layer (alternating Transformer and Mamba blocks) and intra-layer (splitting attention heads within a layer between Transformer and Mamba, then fusing the outputs). The paper compares quality, efficiency, and scaling, and gives design guidance on block ratios and intra-layer choices.

**Strengths:**

+ Clear comparisons and practical guidance: The paper compares inter-layer vs intra-layer hybrids alongside Transformer, Mamba, and sliding-window attention, providing practical design insights.

+ Thorough intra-layer ablations: The paper studies normalization, fusion choices, gating, output projection, and dimension splits for intra-layer hybrid design.

+ Efficiency analysis: The paper shows that lower FLOPs lead to lower step time, higher throughput, and a smaller KV cache than a Transformer at the same scale.

**Weaknesses:**

1. Novelty is limited: Similar hybrid ideas appear in prior work, and they provide a similar recipe and conclusion. Please clarify what is new here beyond existing hybrids and head-wise designs (e.g., what part of the architecture, training recipe, analysis, or design rules is truly novel).

#### references
- Griffin: Mixing Gated Linear Recurrences with Local Attention
- MambaFormer: Hybrid Architecture for In-Context Learning
- An Empirical Study of Mamba-based Language Models
- Jamba: Hybrid transformer-mamba language models
- Jamba-1.5: Hybrid transformer-mamba models at scale
- Hymba: A hybrid-head architecture for small language models

2. Scale: Results focus on 1B-parameter models, 8K training context, and fixed budgets (~60–73B tokens). Several prior hybrid papers (see above references) studied at larger scales (>1B). It would be helpful to demonstrate whether the design rules (e.g., block ratios, placement) still hold for larger models (e.g., 3B–7B) and for longer contexts (e.g., 32K+). Some scaling-law figure would strengthen claims.

3. Hybrid variants not covered: The paper mainly studies head-wise intra-layer fusion, which is already explored in prior work. More experiments with other linear modules (e.g., RWKV, linear attention) or other fusion styles (e.g., sequence-wise fusion) would support the claim that the proposed “best recipe” generalizes across hybrid families.
4. Scope of tasks: The main quality metrics are NLL and a small few-shot set. Downstream long-context tasks (e.g., QA, retrieval, summarization) are limited. Adding these would better validate the method.

**Questions:**

See weaknesses.

---

> ### Author Response · Authors · 2025-11-24
>
> We thank the reviewer for their thoughtful comments. We are happy that they highlighted the clarity of our comparative analysis and the practical insights derived from our thorough intra-layer ablations. We provide responses to the reviewer’s specific questions and concerns (quoted) below.
>
> ---
>
> > **[W1] Limited novelty.**
>
> **[A]** We have detailed the novelty of our work in **Appendix C**. Our primary contributions are as follows:
>
> * **Transparent Design Insights:** We openly share architectural insights regarding optimal design choices for each hybridization strategy, grounded in a comprehensive empirical study.
> * **Comparative Analysis:** We conduct an in-depth comparison of the trade-offs between inter-layer and intra-layer hybrids—two distinct streams that have traditionally been explored in parallel.
>
> While numerous studies on hybrid architectures exist, prior research has predominantly focused on scaling these architectures into foundation models or optimizing data and training pipelines. Crucially, the empirical evidence and rationale behind their specific architectural recipes have often remained undisclosed.
>
> Exceptions are limited: Jamba [1, 2], Empirical [3], RWKV-X [4], and Kimi-Linear [5] compare only a few block ratios, while Empirical [3] and Samba [6] examine positioning of FFN or self-attention blocks in a few configurations. Similarly, Hymba [7] reports results for only a single concatenation variant, and Falcon-H1 [8] directs its deep ablations primarily toward the Mamba component itself rather than the broader hybridization strategy (though both explore parameter ratios).
>
> Furthermore, a fair and rigorous comparison between these two main hybrid paradigms has been missing, and comparisons with homogeneous intra-models, such as differential architectures [9, 10], have rarely been conducted properly.
>
> Therefore, this work aims to provide optimized recipes for diverse hybridization methodologies and openly share our comparative findings to facilitate the future development of hybrid architectures.
>
> ---
>
> > **[W2] Validation at larger scales.**
>
> **[A]** We acknowledge that many prior works on hybrid architectures have prioritized larger model scales, training with more tokens, and context lengths extending well beyond 8k. Since their primary objective is often the development of large-scale foundation models, they typically address the full development pipeline, including extensive data construction and training pipeline optimization including post-training phase.
>
> In contrast, the primary goal of our work is to granularly identify the optimal design choices for each hybridization strategy and to provide a rigorous comparison of these approaches in terms of both model quality and efficiency. This objective required extensive ablation studies; therefore, we focused primarily on the 1B parameter scale with an 8k sequence length and 60B training tokens. We note that even this controlled setting is computationally intensive, requiring approximately 380 H200 GPU hours per experiment.
>
> Nevertheless, to validate our findings across scales, we compared the performance of four model types—Transformer, Mamba, Inter-layer hybrid, and Intra-layer hybrid—using their best-performing configurations (refer to **Appendix H** for scaling law experiments). As shown in **Table R4-1**, hybrid architectures consistently outperformed homogeneous models.
>
> **Table R4-1. Performance comparison at 3B scale.**
>
> | Budget | Models | Token | NLL |
> | :--- | :--- | :---: | :---: |
> | **2e19** | Llama | 1B | 3.554 |
> | | Mamba | 1B | 3.367 |
> | | Inter-H | 1B | 3.403 |
> | | Intra-H | 1B | 3.425 |
> | **4e19** | Llama | 2B | 3.275 |
> | | Mamba | 2B | 3.133 |
> | | Inter-H | 2B | 3.147 |
> | | Intra-H | 2B | 3.156 |
> | **8e19** | Llama | 4B | 3.212 |
> | | Mamba | 4B | 2.962 |
> | | Inter-H | 4B | 2.962 |
> | | Intra-H | 4B | 2.968 |
> | **2e20** | Llama | 10B | 2.884 |
> | | Mamba | 11B | 2.790 |
> | | Inter-H | 11B | 2.777 |
> | | Intra-H | 11B | 2.782 |
> | **4e20** | Llama | 19B | 2.773 |
> | | Mamba | 21B | 2.691 |
> | | Inter-H | 22B | 2.672 |
> | | Intra-H | 21B | 2.676 |
>
> However, it is important to note that FLOPs budgets up to 4e20 (corresponding to a relatively lower token count of approximately 20B) tend to favor Mamba-based architectures. As illustrated in **Figure 7** of **Appendix H**, Mamba’s optimal parameter size for the 4e20 budget is around the 3B scale. Due to this regime-specific advantage, the pure Transformer appears comparatively less effective.
>
> Thereby, within this specific compute budget, the Inter-layer and Intra-layer hybrids exhibit similar performance levels, with neither showing a distinct advantage over the other. We anticipate that the performance trends we observed will persist at larger training budgets.

---

> > ### Author Response · Authors · 2025-11-24
> >
> > > **[W3] Hybrid variants not covered.**
> >
> > **[A]** We acknowledge the extended evaluation to diverse linear attention mechanisms will better support our conclusions. We are currently addressing this by running additional experiments and will notify you as soon as the results are available.
> >
> > Sequence-wise fusion remains a relatively underexplored domain, with TransMamba [11] being one of the few representative works. This approach involves switching between global attention and Mamba components based on sequence position. The primary challenges in this domain—such as identifying the optimal transition point and designing effective memory converters—represent a distinct line compared to our work on inter- and intra-layer hybrids, which focuses on architectural variants, block ratios, and layer positioning.
> >
> > Nevertheless, we believe that investigating optimal strategies for sequence-wise fusion, or exploring its potential synergy when combined with inter- or intra-layer hybridization, constitutes another valuable research direction.
> >
> > ---
> >
> > > **[W4] Limited scope of tasks.**
> >
> > **[A]** We agree that evaluating the model only on language modeling and retrieval does not fully demonstrate its advantages. We are currently benchmarking fine-tuning performance on QA, summarization, and reasoning tasks. Since a 60B-token pre-training scale naturally limits zero-shot performance on complex tasks (e.g., math and code), our evaluation focuses on the model's adaptability via fine-tuning.
> >
> > ---
> >
> > **References:**
> >
> > [1] Lieber, et al. “Jamba: A Hybrid Transformer-Mamba Language Model”. ICLR 2025.
> > [2] Jamba team. “Jamba-1.5: Hybrid Transformer-Mamba Models at Scale”. arXiv 2024.
> > [3] Waleffe, et al. “An Empirical Study of Mamba-based Language Models”. arXiv 2024.
> > [4] Hou et al. “RWKV-X:ALinear Complexity Hybrid Language Model”. arXiv 2025.
> > [5] Kimi team. “Kimi Linear: An Expressive, Efficient Attention Architecture”. arXiv 2025.
> > [6] Ren et al. “Samba: Simple Hybrid State Space Models for Efficient Unlimited Context Language Modeling”. ICLR 2025.
> > [7] Dong et al. “Hymba: A Hybrid-head Architecture for Small Language Models“. ICLR 2025.
> > [8] Falcon team. “Falcon-H1: A Family of Hybrid-Head Language Models Redefining Efficiency and Performance”. arXiv 2025.
> > [9] Ye, et al. “Differential Transformer”. ICLR 2025.
> > [10] Schneider, et al. “Differential Mamba”. AACL 2025.
> > [11] Li, et al. “TransMamba: Flexibly Switching between Transformer and Mamba”. arXiv 2024.

---

> > ### Comment · Reviewer_xUqE · 2025-11-27
> >
> > Thank you for the response. This clarifies several of my concerns. I appreciate that additional experiments are underway, as they will help address other remaining weaknesses. Please share the new results as soon as they are available.

---

> > > ### Author Response · Authors · 2025-11-30
> > >
> > > We appreciate your valuable time and constructive comments.
> > >
> > > Regarding **[W4]**, we have updated the manuscript to include downstream task experiments in **Appendix L**, covering three summarization and three question-answering benchmarks. While Mamba and SWA demonstrated competitive performance in language modeling, they suffered significant performance degradation on these downstream tasks. In contrast, hybrid architectures consistently achieved superior or comparable performance to the baselines across all evaluated tasks.
> > >
> > > Regarding the additional hybrid variants, we will post a follow-up comment if the results become available before the rebuttal deadline.

---

### Official Review · Reviewer_wjS1 · 2025-10-31

**Soundness:** 3
**Presentation:** 3
**Contribution:** 3
**Rating:** 6
**Confidence:** 2

**Summary:**

ChatGPT said:

The paper explores hybrid large language model architectures that integrate self-attention layer with state-space modules to optimize the trade-offs between model accuracy, computational efficiency, and long-context performance. Two main hybridization strategies are examined: inter-layer hybridization, which alternates self-attn blocks and Mamba blocks sequentially, and intra-layer hybridization, which fuses both mechanisms within a single layer through head-wise or sequence-wise splitting. Experiments conducted under controlled compute and data budgets (1B parameters, 8K context length, 4.5e20 FLOPs) compare these hybrids against baselines like Llama, SWA, and Mamba across language modeling quality, efficiency, long-context retrieval, and scaling behavior (including Mixture-of-Experts integration). Results show that both inter- and intra-layer hybrids outperform pure Transformer and Mamba models in perplexity and accuracy, with intra-layer hybrids achieving the best quality-efficiency balance. The hybrid models preserve Mamba’s linear scalability and cache efficiency while retaining the Transformer’s representational power, leading to strong long-context extrapolation and superior performance on benchmarks. Optimal configurations feature approximately a 1:5 Transformer-to-Mamba block ratio and Transformer placement in middle layers, while the addition of Mixture-of-Experts modules further enhances performance and scalability.

**Strengths:**

1. Thorough and systematic investigation: Rather than merely introducing a new hybrid model, the paper carefully examines the design space—including block ratios, layer placement, and fusion strategies.
2. Meaningful insights for practitioners and following works: The design recommendations (e.g., middle Transformer placement, 1:5 ratio) provide actionable guidelines for future hybrid model development.
3. The study shows that hybrid architectures integrate smoothly with Mixture-of-Experts (MoE) layers and retain strong generalization performance—a notable result given the growing adoption of MoE designs in modern large language models.

**Weaknesses:**

Overall, this is a strong and well-executed paper, and the notes below are not weaknesses but rather some of my concerns/questions for further exploration.

1. Limitations of Equal-FLOP Comparisons: While equal-FLOP evaluations provide a fair computational baseline, they may overlook important nuances in training behavior—such as differences in convergence speed, or stability across architectures. Future work could benefit from analyzing learning curves and optimization dynamics to better capture these distinctions.

2. Narrow Task Scope and Generalization: The study focuses primarily on language modeling (ppl) and retrieval task, without assessing reasoning, code generation, or multimodal tasks. This leaves some uncertainty about the conclusion of this paper to broader domains and complex cognitive tasks. Evaluating such capabilities would strengthen the paper’s claims about overall versatility and robustness.

**Questions:**

See Above

---

> ### Author Response · Authors · 2025-11-24
>
> We thank the reviewer for their constructive feedback and are pleased that they valued our systematic investigation of the hybrid design space, the actionable guidelines provided for practitioners, and our demonstration of seamless integration with MoE architectures. We provide responses to the reviewer’s specific questions and concerns (quoted) below.
>
> ---
>
> > **[W1] Limitations of equal-FLOP comparisons.**
>
> **[A]** We appreciate your valuable feedback. We plan to revise the **Appendix** to include the training curve analysis and will notify you once updated.
>
> Regarding our findings, the Transformer showed faster initial convergence than Mamba. However, both architectures eventually converged to comparable training loss levels (under 60B token budgets).
>
> Furthermore, we observed that both computation primitives demonstrated stability during the training phase. The Hybrid model exhibited rapid convergence even in the early stages and demonstrated faster performance gains compared to homogeneous architectures in middle phases.
>
> ---
>
> > **[W2] Narrow task scope and generalization.**
>
> **[A]** We agree that limiting the evaluation to language modeling and retrieval isn't enough to support the Hybrid model's superiority.
>
> We are currently evaluating fine-tuning performance on downstream tasks such as QA, summarization, and math reasoning.
>
> Since the model was pretrained on only 60B tokens, it naturally struggles with tasks like math reasoning and code generation in a direct zero-shot setting, so we are focusing on its fine-tuned performance instead.

---

### Official Review · Reviewer_bLdq · 2025-10-31

**Soundness:** 3
**Presentation:** 3
**Contribution:** 2
**Rating:** 2
**Confidence:** 5

**Summary:**

This work investigates hybrid language models through systematic comparisons of different hybridization strategies and analyses of the key factors underlying their effectiveness. It compares inter- and intra-layer hybridization (and their variants) with pure models, benchmarking them on commonsense reasoning and retrieval tasks. The study finds that intra-layer hybridization is the most promising design choice and provides design insights through ablation studies.

**Strengths:**

1. The research question explored in this work, i.e., investigating optimal hybridization strategies, is important, especially given the growing interest in hybrid model architectures.

2. The study finds that intra-layer hybridization is the most promising design choice, and the ablation study provides valuable reference for the community.

3. The paper is clearly written and easy to follow.

**Weaknesses:**

1. The major concern of this work lies in the generality of its conclusion regarding the ranking between intra- and inter-layer hybrid models, i.e., whether the observed differences are due to the stacking strategy itself or influenced by other design factors.

More specifically, the following evidence suggests that the latter may have an even greater impact:

(1) The intra-layer results in Table 5 also show some ambiguity. For instance, while the authors mentioned that normalization is critical, Hymba w/o group normalization achieves +1.5% higher accuracy than Hymba. In addition, other design choices also show mixed results. For example, it is unclear which fusion strategy, add or diff, is better, as they exhibit different rankings when out=1 and 2. It raises the question of whether the higher accuracy reflects genuinely better design choices or merely fluctuations. If it is the latter, it becomes difficult to claim that intra-layer designs will consistently outperform inter-layer designs across different model and data scales.

(2) Furthermore, from the intra-layer results in Table 5, before tuning the design choices, the vanilla intra-layer model achieves around 53.9% accuracy, while the best accuracy for inter-layer models is 54.0% in Table 4. It is not entirely convincing why the authors chose to report 53.3% for inter-layer models in Table 2. This suggests that intra-layer models require additional architectural tuning to surpass inter-layer models, and the latter could also benefit from tuning, e.g., by adding extra normalization layers.

2. Building upon the previous concern, another major issue is that the authors only report accuracy values with relatively small differences, without providing insight into why each design choice works. Without such analysis, it is difficult to determine which design choices should be preferred for different model or data scales, especially given the mixed results of strategies such as fusion and normalization.

3. The authors mention that the final model choices in Table 2 are also based on efficiency. However, a missing component is the accuracy–efficiency frontier across model variants, which would help reveal which variants offer the best scaling.

4. Almost all of the explored design choices have been discussed in prior works, and some conclusions have already been reported (e.g., Hymba also found that intra-layer stacking is better than inter-layer stacking). Without analyzing the underlying mechanisms behind each design choice, the technical novelty of this work remains limited.

5. Only Mamba is considered as the hybrid operator. Given the emergence of linear attention mechanisms with stronger recall capabilities, it remains unclear whether the conclusions drawn in this work can be generalized to a broader range of hybrid models.

**Questions:**

I have listed my questions in the weakness section. I'm willing to adjust my scores if my concerns are properly addressed.

---

> ### Author Response · Authors · 2025-11-24
>
> We thank the reviewer for their careful review, and we are glad that they found our comparative analysis thorough, our empirical demonstration of efficiency valuable, and the investigation into optimal hybridization strategies important and clearly presented. We provide responses to the reviewer’s specific questions and concerns (quoted) below.
>
> ---
>
> > **[W1-1] Doubt on generality of conclusion.**
>
> **[A]** Thank you for raising this valuable concern. First of all, as you know, few-shot accuracy exhibits significant fluctuation (models with much higher NLL can sometimes show much higher spikes in performance on specific few-shot tasks). Given that we had to experiment with a wide variety of architectural variants, and each model was trained on 60B tokens (which took almost 380 GPU hours in H200 even for the 1B scale), achieving robustness in few-shot accuracy was challenging.
>
> Therefore, we primarily relied on NLL (Negative Log-Likelihood), which has lower variance, to determine the best-performing architecture. We believe NLL serves as a more robust metric for comparison since all models share the same context length, learning rate, scheduler, and optimizer.
>
> To aid understanding of the Intra-layer hybridization, we have added a detailed explanation of the variants we explored in **Appendix E** and included comprehensive ablation results in **Appendix I**. The conclusions drawn from these experiments are summarized at the end of **Table 10** and **11**. Although minor differences exist across model scales, the conclusions are consistent (please refer to NLL more closely), and the resulting best-performing architecture demonstrated strong performance when validated up to the 3B scale.
>
> The architectural changes that showed consistent conclusions in performance improvement include:
> * **Group normalization**: was consistently a variant that significantly reduced NLL across various combinations.
> * **Not using a scaling factor**: was better for performance.
> * **The Fusion operation itself**: had little impact on performance. Since Mamba's implicit attention scores can be negative, the 'add' and 'difference' operations in the fusion step are functionally very similar in their equations (especially when a scaling factor is jointly learned, there is no difference).
> * **Using separate output projection**: before fusion contributed the most to performance improvement.
>
> Although we attempted to keep the parameter count nearly equal for a fair comparison between variants, slight variations still occurred, which might introduce some external factors. However, based on the results across numerous variants, we believe that the architectural variants we identified represent a generally applicable conclusion.
>
> ---
>
> > **[W1-2] Doubt on potential benefit of inter-hybrid tuning.**
>
> **[A]** The performance results for the inter-layer hybrid models reported in **Table 4** are already the result of optimal tuning regarding the position of the Transformer component. Similarly, **Table 5** presents the initial results for tuning the architectural variants of intra-hybrid models, starting from the 1:0 ratio (where all layers are intra-hybrid blocks). The vanilla model with 53.9% accuracy that you pointed out is simply one variant attempted during this tuning process. Like the inter-hybrid model, we sequentially tuned the search spaces like design choices, dimension ratio, and block position. And then, we ultimately used the best-performing structure for each respective strategy in **Table 2**.
>
> Furthermore, **Table 2** compares both hybridization methods using a 1:5 block ratio (considering efficiency relative to the homogeneous model). This is an extremely fair setting: the total number of layers is 13 in both cases, with 11 Mamba modules placed in the same block positions, and only two layers are switched between a pure Transformer module and an intra-hybrid module. We did not unfairly set something to degrade the performance of the inter-layer hybrid model.
>
> The structure of the inter-hybrid model is inherently very simple, so there are few separate architectural variants to explore. The normalization layers within them just follow the already optimized Transformer and Mamba modules, and we are unsure if an extra normalization would be helpful (and we are also unclear where exactly it should be augmented). Notably, the reason we included a normalization layer in the Intra-hybrid module was strategic: since the output scales of the two internal modules differ by a factor of 10 or more, we hypothesized that scaling the outputs before fusion would be more effective for the model to properly learn the relative contributions of both modules.

---

> > ### Author Response · Authors · 2025-11-24
> >
> > > **[W2] Insights why each design choice works.**
> >
> > **[A]** We also agree it is crucial to demonstrate a robust ablation for intra-layer hybridization, given the numerous possible variants. Accordingly, we conducted a wide range of experiments. We have further summarized the design axes we searched across in **Appendix E**, which we encourage the reviewer to refer to.
> >
> > We would like to shortly elaborate further on fusion and normalization here. We consider normalization essential because the output scale of the Mamba component was typically more than 10 times larger than that of the Global Attention component. Although the output projection weight could eventually adjust this difference, we believe that matching the scales is far more beneficial for more stable optimization.
> >
> > In the case of the Differential Transformer, the attention scores of both modules are always positive, which resulted in a significant performance difference depending on whether the fusion operation was addition or difference (see rows 3 and 4 in **Tables 10** and **11**).
> > However, since the Mamba's implicit attention scores can naturally be negative, these operations (add vs. diff) did not yield a statistically significant difference in performance.
> >
> > Similarly, Concatenation is functionally equivalent to the addition of outputs before a separate output projection (though this depends on the exact placement of normalization). Therefore, we narrowed our options down to three candidates (highlighted by gray). We ultimately chose a structure (the final diff-style fusion operation) that is more suitable for expert parallelism and kernel operations.
> >
> > ---
> >
> > > **[W3] Missing accuracy-efficiency frontier.**
> >
> > **[A]** We have already plotted the accuracy-efficiency Pareto frontier in **Figure 1b**. We compared the NLL performance and inference throughput for five model variants. For this comparison, the hybrid models used their respective tuned final architectures, and we reported the frontier across six block ratios, ranging from 1:0 to 0:1.
> >
> > While the Inter-layer hybrid approaches a pure Transformer at the 1:0 ratio, the Intra-layer hybrid remains a hybrid architecture even at this extreme case. Consequently, unlike a U-shaped frontier line, we observe that the Intra-layer hybrid achieves significantly better model quality at a superior inference speed, even at these extreme configurations.
> >
> > We believe this Pareto frontier offers valuable insight for comparing the two hybridization strategies. We will update the detailed values for each setting shortly.
> >
> > ---
> >
> > > **[W4] Technical novelty of this work.**
> >
> > **[A]** Thank you for highlighting this good point. We have elaborated on the novelty of our work in **Appendix C**. Our key contributions are as follows:
> >
> > * Openly sharing architectural insights on the optimal design choices for each hybridization strategy, grounded in a comprehensive study.
> > * Conducting an in-depth comparison of the trade-offs between inter-layer and intra-layer hybrids—two distinct streams that have traditionally been researched in parallel.
> >
> > Although many works on inter- and intra-hybridization have been released, existing research has predominantly focused on scaling hybrid architectures into foundation models and optimizing data composition and training pipelines. Crucially, the experimental evidence and insights behind their specific architectural recipes have often not been fully disclosed or transparent.
> >
> > There are only a few exceptions: Jamba [1, 2], Empirical [3], RWKV-X [4], and Kimi-Linear [5] compare a limited number of block ratios while Empirical [3] and Samba [6] investigate positioning of FFN or self-attention blocks in a few cases. Similarly, Hymba [7], for example, shares results for only a single concatenation variant, and Falcon-H1 [8] focuses its deep ablations primarily on the Mamba component itself rather than the overarching hybridization strategy (although both works did ablation studies on parameter ratios between two modules).
> >
> > Furthermore, a fair and rigorous comparison between the two main hybrid approaches has been missing, and even comparisons with homogeneous intra-models, such as differential architectures [9, 10], have not been properly conducted.
> >
> > Therefore, with this work, we aim to provide optimized recipes for various hybridization methodologies and openly share our comparative findings with the community to aid the future development of hybrid architectures.
> >
> > ---
> >
> > > **[W5] Broader range of hybrid operators.**
> >
> > **[A]** We also agree that validation across a broader range of linear attention mechanisms is necessary to make our claims more convincing. This is currently a work in progress, and we will notify you as soon as the experimental results are ready.

---

> > > ### Author Response · Authors · 2025-11-24
> > >
> > > **References:**
> > >
> > > [1] Lieber, et al. “Jamba: A Hybrid Transformer-Mamba Language Model”. ICLR 2025.
> > > [2] Jamba team. “Jamba-1.5: Hybrid Transformer-Mamba Models at Scale”. arXiv 2024.
> > > [3] Waleffe, et al. “An Empirical Study of Mamba-based Language Models”. arXiv 2024.
> > > [4] Hou et al. “RWKV-X:ALinear Complexity Hybrid Language Model”. arXiv 2025.
> > > [5] Kimi team. “Kimi Linear: An Expressive, Efficient Attention Architecture”. arXiv 2025.
> > > [6] Ren et al. “Samba: Simple Hybrid State Space Models for Efficient Unlimited Context Language Modeling”. ICLR 2025.
> > > [7] Dong et al. “Hymba: A Hybrid-head Architecture for Small Language Models“. ICLR 2025.
> > > [8] Falcon team. “Falcon-H1: A Family of Hybrid-Head Language Models Redefining Efficiency and Performance”. arXiv 2025.
> > > [9] Ye, et al. “Differential Transformer”. ICLR 2025.
> > > [10] Schneider, et al. “Differential Mamba”. AACL 2025.

---

> > > > ### Comment · Reviewer_bLdq · 2025-11-27
> > > > **Further response**
> > > >
> > > > Thank you to the authors for providing the author response. However, the following concerns remain:
> > > >
> > > > [W1-1] Doubts on the generality of the conclusion due to mixed results
> > > >
> > > > I understand that commonsense reasoning accuracy naturally exhibits high variance, as the authors mentioned. However, I do not fully agree that NLL alone can serve as a reliable metric for determining model architectures. Many expected abilities like recall and retrieval are not captured by NLL scores. Although finding a golden metric for LM evaluation remains an open problem, relying solely on NLL while neglecting accuracy-based metrics may not lead to proper design choices.
> > > >
> > > > [W1-2] Doubts on the potential benefit of inter-hybrid tuning
> > > >
> > > > I acknowledge that the authors have done a solid job tuning the common design factors shared by both inter- and intra-layer strategies. However, design factors specific to intra-layer designs, e.g., the additional normalization layers, were not tuned for the inter-layer variants. Since the authors showed that varying these extra factors could change the ranking and conclusions, it is important to design experiments or provide qualitative analysis to clarify the impact of these extra design factors.
> > > >
> > > > This is also connected to concerns regarding the insights into why each design choice works. If quantitative or qualitative analysis/visualization were provided for each design factor (rather than analysis based purely on NLL), it would help clarify the role of each component and whether the improvements truly stem from the stacking strategy.
> > > >
> > > > [W4] Technical novelty of this work
> > > >
> > > > As the authors mentioned, the major contribution is to incrementally vary certain design factors in existing architectures and benchmark them under a fair setting. While this is valuable, without deeper insights or analysis on the effectiveness of each component and without presenting a new conclusion (as prior works have already found that intra-layer strategies perform better), it is difficult to assess the true contribution of this work to the community.
> > > >
> > > > Given the concerns above, I will maintain my current rating.

---

> > > > > ### Author Response · Authors · 2025-11-30
> > > > >
> > > > > We respect your decision. We would like to offer some final clarifications regarding the remaining concerns.
> > > > >
> > > > > **[W1-1]** We agree that NLL alone is insufficient to fully capture model capability; this is precisely why we validated our hybrid models using diverse retrieval benchmarks and the newly added downstream tasks. We primarily relied on NLL for the ablation study solely due to the expensive computational cost of fully evaluating all 45 intra-hybrid variants.
> > > > >
> > > > > However, as shown in **Appendix H**, we did also consider few-shot accuracy to ensure our selection was robust. We adopted group normalization because it demonstrated consistent improvements in both NLL and few-shot accuracy (although there was a specific outlier as in Table 5). We do not believe our findings make mixed results that bring doubt on generalizability.
> > > > >
> > > > > ---
> > > > > **[W1-2]** To ensure a fair comparison, we optimized the design factors specifically tailored to each strategy. The additional normalization, fusion operations, and scaling strategies were design considerations necessitated specifically by the 'intra' hybrid architecture.
> > > > >
> > > > > Regarding your suggestion for Inter-hybrid architectures, it remains unclear where additional normalization or specific tuning could be effectively applied within the existing pre-norm framework. Inserting additional normalization into the residual connection is possible, but likely of marginal utility. While we are open to exploring this in future experiments, it is currently difficult to identify specific tuning strategies that would exclusively benefit the Inter-hybrid approach (without applying to Intra-hybrid as well).
> > > > >
> > > > > As previously mentioned, conducting an in-depth analysis for all 45+ variants was computationally prohibitive; thus, we relied on NLL and few-shot accuracy as primary indicators. However, since we strictly controlled all other variables—such as parameter count and training settings—we believe our ablation study remains robust and reliable.
> > > > >
> > > > > ---
> > > > > **[W4]** While we acknowledge that the Hymba paper also identifies the superiority of intra-layer strategies, we firmly believe that arriving at a similar high-level conclusion does not diminish the contribution of our work.
> > > > >
> > > > > The Hymba paper primarily focuses on proposing a specific high-performance model. While the introduction of multi-head parallel architectures to enhance sequential fusion stands as a primary contribution, the detailed design recipe—beyond a brief ablation on parameter ratios—does not appear to be fully disclosed. Instead, the paper places greater emphasis on the comprehensive pipeline and efficiency techniques, such as KV sharing and Meta-tokens.
> > > > >
> > > > > In contrast, our work focuses on deeply analyzing the hybrid architectures themselves—an area that previous works did not fully share. By providing a broad and rigorous comparison among two hybridization models, SWA, and homogeneous models, we believe our study offers distinct and valuable insights to the research community.

---

### Official Review · Reviewer_FVoW · 2025-11-01

**Soundness:** 3
**Presentation:** 2
**Contribution:** 2
**Rating:** 4
**Confidence:** 5

**Summary:**

The paper compares inter-layer and intra-layer Transformer-Mamba hybrid models across language modeling perplexity, few-shot tasks, long-context retrieval and efficiency, under equivalent number of FLOPs. In the 1B-parameter scale models, hybrid architectures outperform Transformer, SWA or Mamba, and show sub-quadratic memory and throughput scaling. Notably, the hybrid architectures also enjoy the length extrapolation gains from the Mamba component. The analysis provides some design guidance: quality peaks at 1:1 and quality/efficiency Pareto is achieved at 1:5, and etc. The method is also MoE capable and exhibits scaling.

**Strengths:**

- Thorough ablation/analysis of the model design, and the performance comparison with fixed number of FLOPs. The efficiency of hybrid models, in FLOPs, memory, and step-time, is evidenced through experiments beyond theory.
- The design insights (quality at 1:1, Pareto optimal at 1:5) and placement rules, such as not putting Transformer at the front for inter-layer hybrid. For intra layer hybrid, norm, fusion and projection choices are pinned.
- Scaling and MoE compatibility

**Weaknesses:**

Some minor weaknesses and questions are listed here and in the next section.

- Missing citation of https://arxiv.org/abs/2402.04248 - one of the first works to investigate Inter-layer Hybrid Architectures and Positional Embeddings. This work also explored placing the Mamba layer in the front (while dropping the Positional Embeddings), which could be heavily related to the takeaway of not placing Transformer blocks at the front.

**Questions:**

- What could be an intuitive way to understand the intra-layer hybrid architecture?
- Shouldn't 'Transformer' in Figure 1 be Multi-Head Attention?
- How sensitive is the intra-layer hybrid architecture to the hyperparameter settings?
- Is it possible to verify the scaling at upto 3~7B scale for the best performing inter- and intra- layer models?
- What are some possible failure modes - i.e. would there be specific tasks under which the hybrid architecture's benefit might diminish, or be inferior?

---

> ### Author Response · Authors · 2025-11-24
>
> We thank the reviewer for their careful review. We are glad that the reviewer found our comparative analysis thorough and valued our empirical demonstration of efficiency. We provide responses to the reviewer’s specific questions and concerns (quoted) below.
>
> ---
>
> >  **[W1] One missing citation.**
>
> **[A]** Thank you for pointing out this crucial work. We have included MambaFormer in our revised manuscript. We acknowledge that MambaFormer was the first to explore inter-layer hybrid architectures and proposed placing a Mamba block at the initial layer to eliminate PE.
>
> While we share the architectural conclusion (placing Mamba first), our motivation and analytical findings differ. Our analysis reveals that early GQA blocks in hybrid models show highly uniform attention distributions. This aligns with prior works [1, 2] suggesting that early attention layers generate "Bag-of-Vectors" representations—aggregating information across the sequence to inject global context.
>
> We hypothesize that this global context initialization conflicts with the subsequent Mamba blocks, where blurred global representation dilutes the sequential distinctiveness Mamba requires. Therefore, we place Mamba first to ensure dense sequential encoding is established before Attention layers perform global retrieval. Moreover, we validate this optimal ordering within a more modern LLM setting (with RoPE) for general language modeling.
>
> ---
>
> > **[Q1] Intuitive way to understand intra-layer hybrid architecture.**
>
> **[A]** For better clarity, detailed descriptions of the hybrid architecture are presented in **Appendix E**, and further ablation results are summarized in **Appendix I**.
>
> Intra-layer hybrid approaches typically adopt a “head-wise splitting” architecture. Here, half of the heads utilize GQA (or MHA) and the other half utilize Mamba blocks, then their outputs are merged. This design can be interpreted as an ensemble of architectures with distinct inductive biases. The model applies different perturbations to the input—via the projection matrices of each head—and aggregates the outputs from these heterogeneous blocks.
>
> More specifically, we hypothesize that the mixing of these two mechanisms helps the model to calibrate the attention scores, attending more precisely to relevant positions. This motivation aligns with recent explorations in differential architectures [3, 4]. While Differential Transformer [3] and Differential Mamba [4] demonstrate a noise-canceling effect using two identical computational primitives, we investigate the efficacy of mixing heterogeneous architectures.
>
> As shown in **Table 5**, we provide an ablation study comparing our approach against these differential models. Our results demonstrate that the hybrid model achieves significantly superior performance, offering new evidence that mixing distinct architectures provides a stronger advantage than homogeneous mixing.
>
> ---
> > **[Q2] Wrong terminology used in Figure 1.**
>
> **[A]** Thanks for the comment. A standard Transformer block consists of Multi-Head Attention (MHA) followed by a Feed-Forward Network (FFN), whereas a Mamba block refers to the SSM-based layer (the original Mamba paper eliminated the FFN part by integrating a gated MLP design into the SSM block). Therefore, as correctly pointed out, the precise primitive in **Figure 1** is indeed MHA (or GQA in our case).
>
> We used the term 'Transformer' rather than global attention (MHA or GQA) just to help intuitive understanding of the model as a hybrid between the Transformer and Mamba architectures. We have clarified this terminology in the caption of the revised manuscript.
>
> ---
>
>
> > **[Q3] Sensitivity analysis for intra-layer hybrid architecture.**
>
> **[A]** Intra-layer hybrid models have consistently shown superior performance compared to others and robustness across a diverse range of hyperparameters, including model size, learning rate scheduling, and the number of training tokens. For instance, in scaling law experiments (see **Appendix H**) conducted across four model scales under five distinct compute budgets, intra-hybrid models consistently outperformed Transformers, Mamba, and Inter-layer hybrid models.
>
> Furthermore, it yields consistent performance gains even with the integration of Mixture-of-Experts (MoE), further validating robustness.
>
> Moreover, the detailed ablation study results across various design spaces have been updated in **Appendix I**. The results consistently showed reasonable performance trends from the 1B and 350M models trained on 60B tokens.

---

> > ### Author Response · Authors · 2025-11-24
> >
> > > **[Q4] Verifying the best performing hybrid models at larger scales up to 3~7B.**
> >
> > **[A]** We compared the performance of four model types—Transformer, Mamba, Inter-layer hybrid, and Intra-layer hybrid—with their best-performing configuration (in **Appendix H**; scaling law experiments). We found that the hybrid architectures consistently outperformed the homogeneous models.
> >
> > **Table R1-1. Performance comparison at 3B scale.**
> >
> > | Budget | Models | Token | NLL |
> > | :--- | :--- | :---: | :---: |
> > | **2e19** | Llama | 1B | 3.554 |
> > | | Mamba | 1B | 3.367 |
> > | | Inter-H | 1B | 3.403 |
> > | | Intra-H | 1B | 3.425 |
> > | **4e19** | Llama | 2B | 3.275 |
> > | | Mamba | 2B | 3.133 |
> > | | Inter-H | 2B | 3.147 |
> > | | Intra-H | 2B | 3.156 |
> > | **8e19** | Llama | 4B | 3.212 |
> > | | Mamba | 4B | 2.962 |
> > | | Inter-H | 4B | 2.962 |
> > | | Intra-H | 4B | 2.968 |
> > | **2e20** | Llama | 10B | 2.884 |
> > | | Mamba | 11B | 2.790 |
> > | | Inter-H | 11B | 2.777 |
> > | | Intra-H | 11B | 2.782 |
> > | **4e20** | Llama | 19B | 2.773 |
> > | | Mamba | 21B | 2.691 |
> > | | Inter-H | 22B | 2.672 |
> > | | Intra-H | 21B | 2.676|
> >
> >
> > However, we note FLOPs budgets up to 4e20 (relatively smaller token count, approximately 20B) inherently favored Mamba (as shown in **Figure 7** in **Appendix H**, where Mamba’s optimal parameter size for 4e20 budget is around 3B scale). Due to this advantage, the performance of the pure Transformer model was slightly suppressed, and also the Inter-layer and Intra-layer hybrids show similar performance without a distinct advantage over one another. We anticipate that the performance trends we observed will persist at larger training budgets.
> >
> >
> > ---
> >
> > > **[Q5] Possible failure modes of hybrid architectures.**
> >
> > **[A]** We do not anticipate that hybridization will introduce unique failure modes. However, we believe that the hybrid model will fail in extreme cases where its modules severely degrade. For instance, problems could arise in the edge cases of each module, such as a Transformer struggling with length extrapolation or usage in very long contexts, or a Mamba model experiencing degraded recall capabilities.
> >
> > Nevertheless, as shown in **Figure 4**, the hybrid architecture may leverage the strengths of both modules simultaneously to achieve superior performance.
> >
> >
> > ---
> >
> > **References**
> >
> > [1] Clark, et al. “What Does BERTLookAt? AnAnalysis of BERT’s Attention”. ACL 2019.
> > [2] Voita, et al. “Analyzing Multi-Head Self-Attention: Specialized Heads Do the Heavy Lifting, the Rest Can Be Pruned”. ACL 2019.
> > [3] Ye, et al. “Differential Transformer”. ICLR 2025.
> > [4] Schneider, et al. “Differential Mamba”. AACL 2025.

---

### Author Response · Authors · 2025-11-24

Dear Reviewers,

Thank you very much for your valuable time and insightful feedback on our paper. We sincerely apologize for the delay in providing this response.

We have carefully considered all comments and have uploaded a revised manuscript incorporating new experimental results and clarifications to address your main concerns. For your convenience, all revisions have been highlighted in **red**. We kindly ask you to review these updates.

**Specific Updates in the Revised Manuscript:**

1. **Missing References:** We have incorporated the suggested missing references.
2. **Contributions:** We have summarized how our work differentiates itself from prior work in **Appendix C**.
3. **Intra-layer Hybrid Architecture:** A more detailed explanation and the design axes of the Intra-layer hybrid architecture have been added to **Appendix E**.
4. **3B Experiment Details:** The scaling law experiments comparing different architectures at the **3B scale** have been detailed in **Appendix H**.
5. **Intra-Hybrid Ablation Study:** More detailed results for the explored variants of the intra-hybrid model across the **1B** and **350M scales** are now presented in **Appendix I**.
6. **Component Contribution Analysis:** An analysis quantifying the contribution of the **Global Attention** and **Mamba** components in each module has been updated in **Appendix J**.

**Ongoing Experiments:**

We are actively running additional experiments that directly address several points raised in the reviews:

1. **Downstream Task Performance:** We are conducting finetuning experiments on various downstream tasks (**Question Answering**, **Summarization**, and **Reasoning**) to evaluate practical performance.
2. **Other Hybrid Variants:** We are currently implementing and testing additional hybrid variants.
3. **Completing Missing Results:** We are in the process of filling in the remaining experimental data.

We will update the manuscript as soon as these results are available.

Best,
Authors

---

> ### Author Response · Authors · 2025-11-30
>
> Dear Reviewers,
>
> We have updated the manuscript to address your concerns regarding downstream performance and training dynamics. The key updates are as follows:
>
> - **Downstream Task Performance:** In **Appendix L**, we have added fine-tuning results for three summarization and three question-answering downstream tasks.
> - **Training dynamics:** We have included a comparison of learning curves across model architectures in **Appendix K**.
>
> All revisions are highlighted in **red**. We will provide an additional update if the experiments on other hybrid variants are completed before the rebuttal deadline.
>
> Best,
> Authors

---

### Meta-Review · Area_Chair_8LuW · 2026-01-05

**Summary:**

The paper proposes as study on a hybrid architecture designs for LLMs. The paper primarily considers fusion of Mamba and Transformer-Attention layers in inter- and intra-layer fashion. Comparison is done under controlled data and compute budgets. Multiple reviewers had negative reviews with scores 2 (reject), main concerned raised a related to focusing only on Mamba whereas there are multiple other emerging SSM solution: DeltaNet, GatedDelta net etc. Reviewers mentioned limited novelty and that proposed fusions were already proposed in the past. Finally, reviewers mentioned that the evaluation bench is not wide enough: primarily NLL loss is reported, retrieval and few shot tasks at the 1B scale at most.

**Reviewer Concerns:**

The rebuttal addressed missing citations, clarified architectural motivations, added ablations and downstream evaluations. Extra 3B evidence was presented. However, not all evaluations were ready on time, and authors promised to follow up. In summary, reviewers still were not satisfied with reliance on NLL, limited novelty, incomplete isolation of inter- vs intra-layer tuning effects, and generalization beyond Mamba-based hybrids.

**Reviewer Scores:**

Most critical reviewers who scored 2 would not change much:
bLdq: 2 ->  2
xUqE: 2 -> 2,3
Other reviewers will mostly stick to their scores.

---

### Decision · Program_Chairs · 2026-01-26

Reject